# The 2023 Marine Heatwave In The North Atlantic Tropical Ocean

Amélie Loubet[1], Simon J. van Gennip[1], Romain Bourdallé-Badie[1], Marie Drevillon[1]

[1]Mercator Océan International, 2 Av. de l'Aérodrome de Montaudran, 31400, Toulouse, France

*Correspondence to*: Simon J. van Gennip (svangennip@mercator-ocean.fr)

**Abstract.**

In a context of climate change, Marine Heatwaves (MHWs) are becoming more intense, frequent and/or lasting longer. During the year 2023 and based on the Copernicus Marine forecasting system, the Mercator Ocean International MHW bulletin (https://www.mercator-ocean.eu/en/category/mhw-bulletin/) highlighted week after week a MHW event occurring in the North Atlantic (NA) tropical Ocean. In this paper, we propose an 4D characterisation of this event using the Copernicus Marine global reanalyses. We demonstrate how this 2023 MHW event in NA tropical Ocean is extraordinary compared to previous years. All indices commonly used for characterising MHWs (intensity, duration, total activity and area) reached values not observed before at the surface but also in subsurface. The timing of the event and its vertical structure differ across the basin, with the MHW developing first in the North-East, with peaks of intensity in May and progressively moving south westward across the basin. A characterisation of MHWs at all vertical levels reveals that the vertical structure differs across subregions with different processes at play: in the Eastern and subtropical centre of the gyre heat propagates from the surface to the subsurface spanning beyond the mixed layer depth; whereas in the Caribbean region, abnormally warm waters at depth are transported from remote equatorial regions by eddies traversing the area.

**Short summary**

Marine Heatwaves (MHWs) are intensifying due to climate change. In 2023, the Copernicus Marine forecast system tracked a significant MHW event in the North Atlantic Tropical Ocean. Here we show this event was unprecedented, at the surface and at depth. It peaked in the northeast in May, spreading southwest to reach the Caribbean by fall. In the east and centre, the MHW remained within the surface layers, while in the Caribbean, it reached deeper levels due to warm waters advected by equatorial eddies.

# 1 Introduction

The year 2023 was the warmest year on record with annual average global atmospheric temperature reaching $1.43 \pm 0.11$ °C above pre-industrial levels (Foster et al., 2024). Air temperature records were broken for multiple months and regions (WMO, 2024). Europe and the subtropical North Atlantic (NA) region were particularly affected with highest recorded air temperature anomalies (ESOTC, 2023). Abnormally high temperature anomalies have also been detected at the surface of the ocean consistently across products (observation, forecasting system, reanalysis) in the NA where mean temperature estimates have exceeded those of previous years (Copernicus, 2024). A direct result of this warming ocean is the increase of the occurrence of extreme warm events.

When abnormally high ocean temperatures occur for a sustained period of time it leads to an extreme event referred to in the literature as Marine Heatwaves (MHW). A MHW definition was proposed by Hobday at al. (2016, 2018), that has enabled to document in a standardised manner MHW characteristics such as MHW duration, intensity and extent globally. MHW frequency has already increased between 1925-2016 (Oliver et al. 2018) and will keep on increasing due to anthropogenic forcing (Frölicher et al., 2018; Oliver et al., 2019). MHWs threaten marine ecosystems causing harm from species to ecosystem level such as coral bleaching, reduction of habitat-forming seaweed, harmful algal blooms, species range shift and mass mortality events (Le Nohaïc et al., 2017; Wernberg et al., 2013, 2016 ; Smith et al., 2023 ; Cavole et al., 2016).

The regular monitoring of MHW conditions globally (Mercator Ocean International weekly bulletin: https://www.mercator-ocean.eu/en/category/mhw-bulletin/) revealed the prolonged presence across the year of an MHW event within the North Atlantic basin (NA). Studies documenting MHWs in the NA have only been local to regional, with no records of such widespread events occurring (Frölicher and Laufkötter, 2018; Smith et al., 2021; Zhang et al., 2023). Furthermore, MHW have been well studied for the surface where long satellite records exist, but their subsurface extent should be considered more in details (Schaefer et al., 2023; Zhang et al., 2023; Sun et al., 2023). Vertical structure has been studied using in-situ data (El Zahaby and Schaeffer, 2019, 2021; Zhang et al., 2023; Juza et al., 2022; Pirro et al., 2024). Alternative approaches consist in the use of a numerical models (Darmaraki et al. 2019, Sun et al., 2023) which provide a continuous complete 3-dimensional ocean state. In this study, we decided to use an eddy resolving ocean reanalysis (ocean models that use data assimilation) at daily resolution and covering a sufficiently long period to build a 30-year long reliable climatology, as advised by the World Meteorological Organisation (WMO) (WMO, 2018; Hobday et al., 2016, 2018). The regular update of such product to remain close to real time enables to study such recent event, and assess its characteristics relative to previous years.

We propose a 4-dimensional description (3D + time) of the ocean temperature extreme event of 2023 in the NA tropical Ocean using the temperature field of the Copernicus Marine Service GLORYS12V1 reanalysis product (Lellouche et al. 2021), to which Hobday's MHW algorithm has been applied (Hobday at al. 2016, 2018). After the method description in section 2, we propose, in section 3, a characterization of the 2023 event in the NA tropical Ocean, from the surface to the subsurface. Conclusions and perspectives are done in section 4.

**2 Methods**
**2.1 Datasets**

| Product Ref. No. | Product ID & type | Data Access | Documentation |
|---|---|---|---|
| 1 | GLOBAL_MULTIYEAR_PHY_001_030, numerical model | EU Copernicus Marine Service Product (2023) | Product User Manual (PUM): Drévillon et al., 2023a<br>Quality Information Document (QUID): Drévillon et al., 2023b<br>Journal article: Lellouche et al., 2021 |
| 2 | ERA5 reanalysis | Climate data store (https://cds.climate.copernicus.eu) | Hersbach, H., Bell, B., Berrisford, P., Hirahara, S., Horányi, A., Muñoz-Sabater, J., et al. (2020). The ERA5 global reanalysis. Quarterly Journal of the Royal Meteorological Society 146, 1999–2049. doi: 10.1002/qj.3803 |

**Table 1: Product reference table**

The main product used for this study is the GLOBAL_MULTIYEAR_PHY_001_030 reanalysis distributed by Copernicus
Marine Service (https://doi.org/10.48670/moi-00021). This reanalysis is developed from the NEMO global ocean model with
a horizontal resolution of 1/12° (9 km at the equator and 2 km close to the poles) and with 50 vertical levels where observational
products are assimilated using a reduced-order Kalman filter. Along track altimeter data (Sea Level Anomaly), Satellite Sea
Surface Temperature, Sea Ice Concentration and in situ Temperature and Salinity vertical Profiles are jointly assimilated.
Moreover, a 3D-VAR scheme provides a correction for the slowly-evolving large-scale biases in temperature and salinity. This
reanalysis covers the period 1993-onward. It was driven by the ERAinterim atmospheric fluxes from 1993 to 2019, and ERA5
thereafter. A more detailed description and study is proposed by Lellouche et al. 2021. The use of ocean reanalysis makes it
possible to both study surface MHWs and to compare the results with other satellite datasets but also to gain insight in their
vertical structure. This reanalysis is particularly well suited to the study of near-surface phenomena due to its refined vertical
discretization in the first 50 metres of the ocean (first 18[th] layers of the reanalysis). In this study we calculated a 30-year 3D
daily climatology of temperature using the baseline period 1993-2022, and used the data from the year 2023 to characterise
the MHW in the NA tropical Ocean.

**2.2 Characterisation of Marine Heatwaves**

MHWs are prolonged period of abnormally high seawater temperature. We identified an MHW event as a period of at least
five consecutive days where the temperature exceeds the 90th percentile of a 30-year climatology, following Hobday et al.
(2016) recommendations. The 90th percentile and the mean temperature climatology were smoothed using a 31-day moving
window to reduce high-frequency noise while detecting MHWs. First, we detected MHWs for the surface layer in 2023 using
this definition to characterize the studied event. Then, we detected surface MHWs from 1993 to 2022 in order to compare the
MHWs characteristics over the climatology period. Additionally, we detected 2023 MHWs from the surface to 2,225 m depth
(the 41st depth layers of the reanalysis) to investigate subsurface MHW signatures for this particular year.
The detected MHWs were characterised using common metrics such as duration (number of consecutive days above the 90th
percentile threshold), intensity and intensity-based category (moderate, strong, severe and extreme) (Hobday et al., 2016 et
2018). Note that depending on the method, MHW intensity is defined either by the temperature anomaly relative to the mean
climatology (Hobday et al., 2016; Oliver et al., 2018) or relative to the threshold (Darmaraki et al., 2019; Juza et al., 2022).
Here, to focus on the study of extremes, we define MHW intensity as the temperature anomaly relative to the 90th percentile
threshold. We also calculated the annual surface MHW activity (from 1993 to 2023) following Simon et al. (2022) definition:

$$Activity = \sum_{event \subset year} \bar{A}_{event} * d_{event \subset year} * S_{event}$$


where $event$ refers to a specific MHW event, $year$ refers to a specific year, $\bar{A}_{event}$ (in °C) is the temperature anomaly during
the $event$ averaged over its duration, $d_{event \subset year}$ (in days) is the $event$ duration within the specific year and $S_{event}$ (in km²)
is the spatial extent of the $event$. Here we calculated activity for each grid cell, so $S_{event}$ is the surface of the grid cell. Then
we averaged the activity over the studied area to get the annual spatial mean activity (in °C.days.km²).

We defined the studied area to focus on regions with long lasting and intense MHWs, choosing the Atlantic from 10°S to
50°N. We divided the study area into coherent subregions following the definition of the Longhurst biogeochemical provinces
(Reygondeau et al., 2013; Longhurst, 2007; shapefile from Flanders Marine Institute, 2009). Based on the highest mean activity
regions for 2023 (not shown), we focused on the provinces denoted North Atlantic Subtropical Gyral Province (East) (NASE),
North Atlantic Tropical Gyral Province (NATR), and Caribbean Province CARB (Figure 2).

For time series, we spatially averaged the daily MHW intensity and the mixed layer depth (MLD) over each chosen Longhurst
province. To generate the mean vertical MHW intensity profiles for a given province, we first temporally averaged the daily
MHW intensity (using MHW days only) for each grid cell in the province at each depth; then, we spatially averaged the
temporal mean values across all grid cell within the province, at each depth. We thus obtained one spatiotemporally averaged
intensity vertical profile in the given province. We computed the standard deviation of the spatial mean, which provides insight
into the degree of variability or spatial inhomogeneity across the province at each depth. For each province we estimated the
MLD by averaging first temporally (over 2023), then spatially (over each province) the MLD data distributed by
GLOBAL_MULTIYEAR_PHY_001_030.

For horizontal Hovmöller diagrams, daily intensities were spatially selected using a mask with the 3 provinces of focus and
then averaged across latitudes. Thus, when regions overlap in longitude (for instance NASE and NATR), data from both
regions is averaged together. This method was used to generate Hovmöller diagrams for different depth. For the depth/time
Hovmöller diagram, MHW intensity was selected using a mask of the specific region and then averaged over latitude and
longitude.


**2.3 Atmospheric Variables**

Using ERA5 reanalysis air temperature ($TAIR$) and 10m wind ($U10$) data, we computed 2023 anomalies based on 30-year
climatologies (1993-2022) to match the sea temperature climatology baseline (used to detect MHWs).  The air temperature
anomaly ($TAIR - TAIR_{clim}$)  was then smoothed over a 7-day window.  For the wind at 10m, we calculated the anomaly of
the absolute values ($|U10| - |U10_{clim}|$) to focus on anomalies of intensity and not of direction. Then the anomaly was
averaged over 2023.
Daily air temperature anomaly was averaged over latitude and used to generate a Hovmöller diagram (using the same method
than for MHW intensity Hovmöller diagrams, see section 2.2).
**3 Results**
**An event of unprecedented characteristics at the surface**
During the year 2023 an MHW event of extraordinary characteristics occurred in the NA tropical Ocean impacting the entire
ocean region between 10°S and 50°N (Figure 1a). The event developed in March, covering ~20% of the region predominantly
in moderate conditions, to progressively peak from August to mid-October gaining in both extent and intensity occupying over
60% of the area, with strong and higher categories progressively accounting for nearly 60% of the MHW surface by mid-
October (43.5% for strong, 14.8% for severe and 1.1% for extreme on 15th October). A decrease in extent occurred in October,
and in December with in between a small increase in November (Figure 1b).
Overall, nearly the entire area (>99%) has been in MHW conditions at some point across the year, with such conditions going
beyond moderate in terms of category (Figure 2a). Indeed, only 8.3% of the region was exposed to moderate-only events
during the year and corresponds to regions in the vicinity of the Gulf Stream and its extension. In total, 40.2% of the region
has been exposed to a maximum level of category strong, 40.7% of category severe, and 10.8% of category extreme and
beyond. The most intense events span from the Iberian Peninsula, the eastern side of the basin and the Caribbean region.
Noteworthy, the regions with most intense MHW events (Figure 2a) coincide with region with highest number of marine

heatwave days (long lasting MHW areas of Figure 1a); for instance, the region between 15°N and 35°N spanning from the East of the African coast until 40°W and the one close to Hispaniola island in the Caribbean region.

In terms of duration, a large proportion (19.9% of the study region, mostly constrained within the triangle formed by the Iberian Peninsula, western Africa and the Caribbean region) was in MHW condition for more than 250 days during the year (Figure 1a). Most notably, the region off the coast of Morocco was exposed to more than 300 MHW days. The Gulf Stream region was more moderately impacted with around 100 MHW days over the year 2023.

The year 2023 is characterised by an unprecedented MHW event outstanding in all indices commonly used to describe MHWs with highest mean daily intensity, mean duration, total surface exposed and mean activity (Figure 1c). On average over a year, 2023 exceeds all previous years of the reanalysis product (1993-2022) with mean duration of 54 days over the area, mean daily intensity of 0.52 °C and mean activity of 17,204 °C.day-1.km$^2$ (Simon et al., 2022). No other year presents similar high values for a single of these metrics (nevertheless all 3 combined) underlining the extraordinary nature of the 2023 MHW event. Note that this extent corresponds to a strong negative anomaly in surface wind intensity (Figure 1a).

The timing corresponding to the peak of the event in terms of MHW category (Figure 2b) varied geographically, with highest category first reached during springtime in the eastern part of the basin, then during July/August for the centre of the basin, and during fall for the western part.

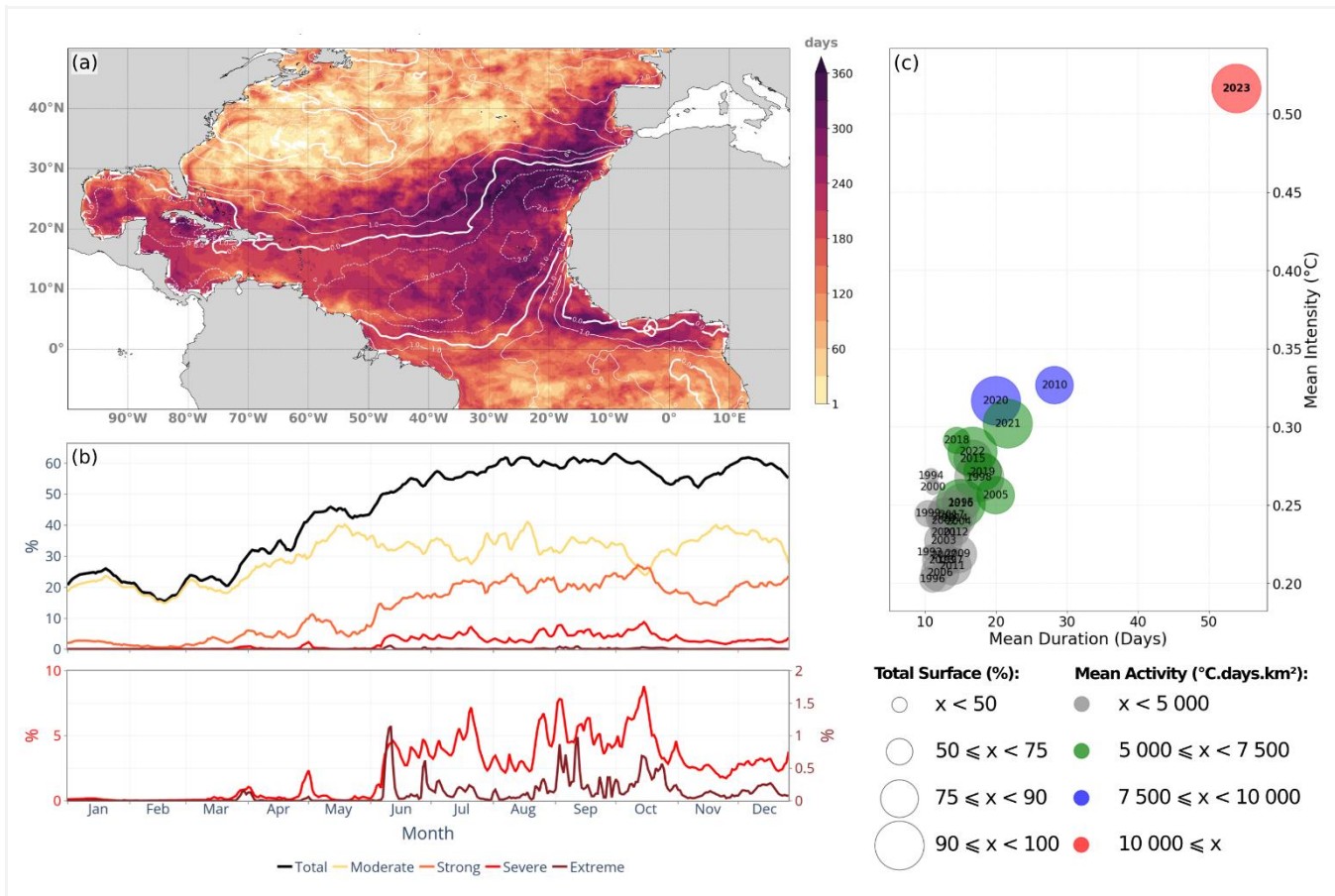

Figure 1: Characteristics of the marine heatwave hitting the North Atlantic across 2023 between 10 °S and 50 °N: total number of heatwave days (panel a); evolution of the total area and area by category affected by MHW events (panel b); representation of the MHW event for 2023 in terms of mean duration, intensity, maximum coverage (bubble size), and activity (coloured bubble) relative to previous years (panel c). White contours in panel a refers to the annual mean of absolute wind anomaly (m/s).







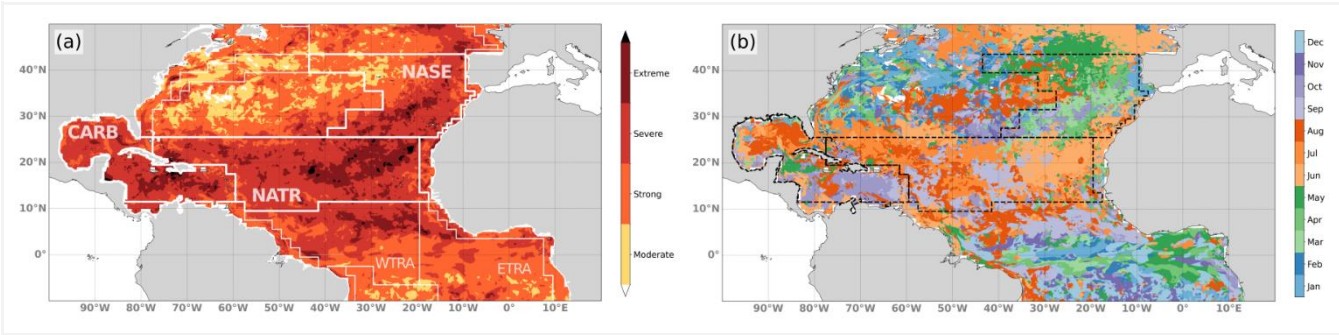

Figure 2: Highest marine heatwave category reached in 2023 (panel a); month during which the highest category first occurred (panel b). Zones delimited in white in panel a and black in panel b refer to the Longhurst bio provinces.

**Regional vertical structure of MHW**
Beyond the extraordinary surface signature of the 2023 MHW event, we further investigate this event by characterising its
vertical structure and evolution over time. For this, we divided the study area into physically coherent subregions as defined
by Longhurst (Reygondeau et al., 2013; Longhurst, 2007) (Figure 2a).
We focused on 3 subregions where intense and long MHW events occurred (Figure 1a, 2a): the North Atlantic Subtropical
Gyral Province (NASE), to the east of the basin; the North Atlantic Tropical Gyral Province (NATR), in the centre; and
Caribbean Province (CARB) to the west. For each subregion, we computed the mean intensity depth profile and estimated the
mean mixed layer depth (MLD) (see Methods).

The depth profile of MHW intensity is not identical across the basin, with significant differences across the region, most
notably for the depth where the maximum intensity occurred (Figure 3a). Intensity peaks at much deeper depth in the CARB
region (max at 156 m, deeper than the mean MLD of 23.8 m represented by red doted horizontal line) than for NASE and
NATR regions. For NASE and NATR regions, maxima occur at 40m and close to the surface, respectively, both within the
mixed layer (MLD represented by blue and green doted horizontal lines). The mean intensity profile of MHWs for NASE
shows homogeneous levels across the mixed layer with slightly higher values at subsurface (40.3 m depth), at the bottom of
the mixed layer. The NATR region shows a different MHW intensity profile than the NASE region, with a maximum in the
surface layer. In addition, we notice from the standard deviation of the MHW intensity (shaded area) that spatial inhomogeneity
is largest for the CARB region for depth between 150 and 400m.

**Evolution of MHW Intensity and Extent Across Depth**
Further insight on MHW characteristics was carried out by evaluating for each region the evolution of intensity and spatial
extent of the MHW at depths where maximum intensity occurred in each region (surface, 40m and 156m) (Figure 3 b, c and
d).

At the surface, like observed in month of highest MHW categories (Figure 2b), later timing in the peak of MHW for the more
westward regions is evidenced in the area averaged intensity. Maximum intensity is reached earlier in the most eastern region,
the NASE region (beginning of May), then in the NATR region (late July), and in the CARB region (October) (Figure 3b solid
blue line, 3c solid green line and 3d solid red lines). We also note a peak in March in CARB region (lasting 10 days and
reaching an intensity of 0.9 °C) which seems to be an isolate event and would require further investigation not done here.

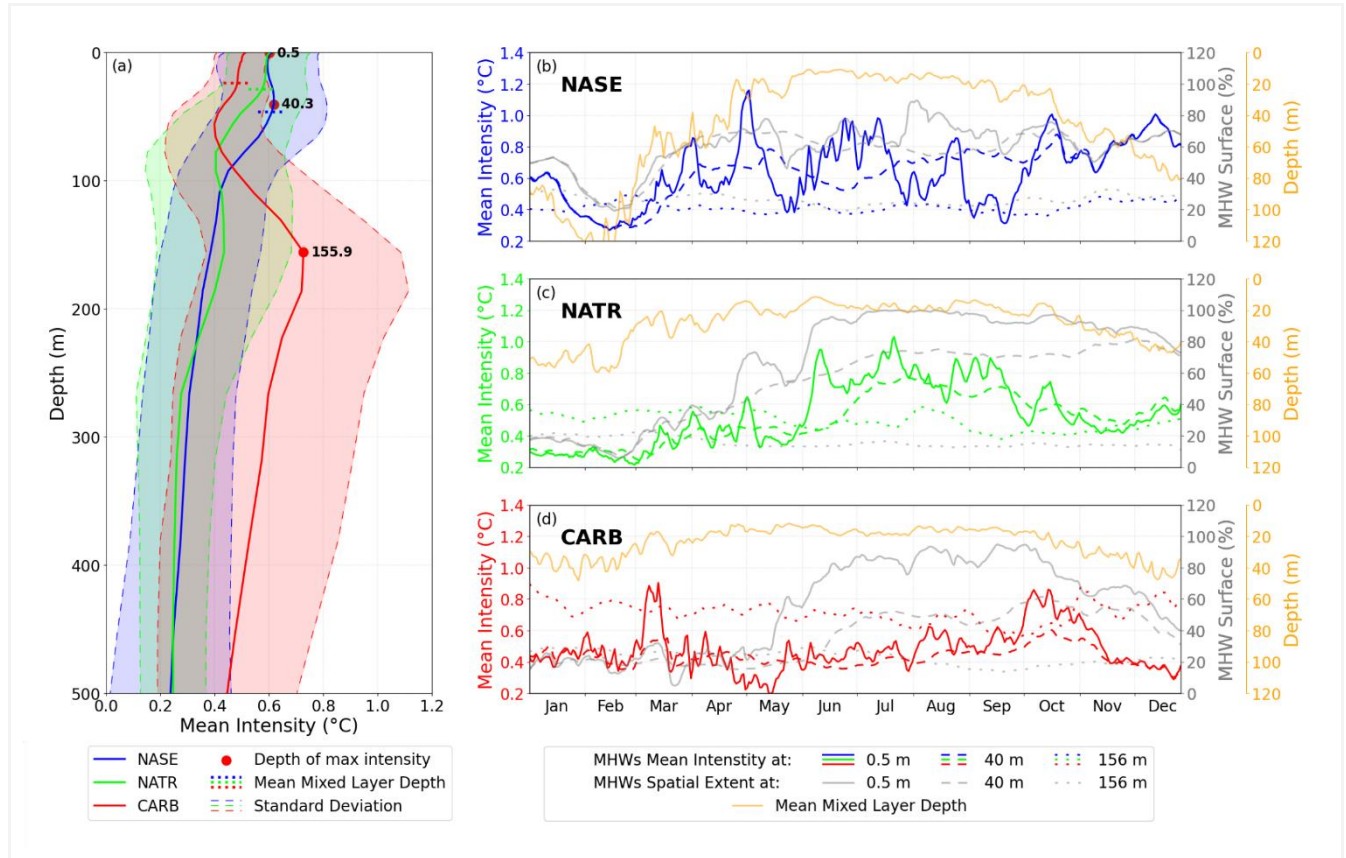

Figure 3: Evolution of the intensity and spatial extent of the 2023 marine heatwave for different regions in the North Atlantic. Mean MHW intensity (in °C) profile (panel a) of NASE (blue), NATR (green) and CARB (red). Shading areas represent standard deviations of spatial mean. Dotted horizontal lines represent the mean MLD and red dots represent depth of highest mean intensity for each region. Time series of mean intensity (in °C), surface coverage (in %) and mean mixed layer depth (in m) in NASE (panel b), NATR (panel c) and CARB (panel d) provinces. Blue, green and red lines (NASE, NATR, CARB respectively) represent the mean intensity at the surface (solid), at 40m (dashed) and at 156m (dotted). Grey lines represent the surface coverage at the surface (solid), at 40m (dashed) and at 156m (dotted). Orange lines represent the mean layer depth in the corresponding region.


For the NASE region, the intensity at the surface shows large fluctuations across the year: in total, six maxima of intensity higher than 0.8°C are observed, followed by low troughs (with differences of 0.3 to 0.8 °C relative to the peaks) (Figure 3b, solid blue line). The intensity at 40 m depth – where maximum occurs in the mean intensity depth profile – increases progressively from 0.3°C early March to peak mid-December at 1.0°C (Figure 3b, dashed blue line). In between, some variations exist with smaller peaks mid-May and mid-October (occurring shortly after the surface peaks). The MHW signature at the surface develops earlier than at subsurface. It suggests that the signal propagates from the surface, across the water column to subsurface, and below the MLD (Figure 3b, orange line). The increase in area occupied by MHWs in March, for both the surface and 40 m depth, coincides with the shoaling of the MLD (Figure 3b, solid and dashed grey lines, orange line). The horizontal extent is similar for both depths, with values fluctuating around 70% of the area from April to mid-October. We note that the intensity at the surface and at 40m depth are equal during winter period. This is linked to the deepening of the MLD to levels deeper than 40m which homogenise temperature (Figure 3b, orange line). Unlike shallower depths, the intensity at 156m remains stable around 0.4°C across the year (Figure 3b, dotted blue line). Extent is lower at 156m depth with values remaining between 20% to 30%. Note, in between 40m and 156m, surface warming propagates progressively at depth across the year. For instance, at 100m depth, from February onwards, the intensity levels steadily increase from values of 0.29°C to 0.61°C by mid-November (see Figure 4e).

The evolution of the mean intensity for NATR, at the surface and at depth, describes a different kind of MHW than for the NASE region (Figure 3c, green lines). The MHW is characterised by one long temporal event – rather than a series of shorter events – that peaks at the end of July. At the surface, high intensity develops rapidly early June and remains high until the end of September with values constantly above 0.5°C (Figure 3c, solid green line). Horizontal extent of the MHW increases in two steps: first reaching ~70% at the end of April-May and then above 90% from mid-June to November, to finally drop slightly below 80% (Figure 3c, solid grey line).

At 40m depth, rapid increase in intensity occurs later relative to the surface starting end of May at 0.4°C to reach a maximum of 0.77°C by the end of July (Figure 3c, dashed green line). Spatial extent increases progressively from ~40% coverage in April to above 80% by the end of November (Figure 3c, dashed grey line). These increases (in intensity and at surface) occurs when the MLD is shallowest (about 20m), meaning that the MHW reaches bellow the MLD (Figure 3c, orange line). At 156m depth, intensity levels vary across the year around 0.4-0.6°C and horizontal extent of MHW remains low and stable (around 15-20%) (Figure 3c, doted green and grey lines). This signal is decorrelated with what is observed for surface layers.

Dynamics for the CARB region differ with the NASE and NATR regions, with an MHW signal at both surface and depth (Figure 3d). At the surface, a late and long-lasting peak of MHW intensity (larger than 0.6°C for 30 days) occur in October (peaks at 0.86°C), after the observed peaks in the other 2 regions (Figure 3b, c and d, coloured solid lines). At 40m depth, intensity levels and fluctuations are similar to the surface, with lower magnitude and reduced high frequency variations (Figure 3d, red dashed line). Timing in the peaks in March and October show a lag relative to the surface. MHW horizontal extent at the surface increases from mid-May (~30%) to peak late September (up to 95% of the area), to then decrease by the end of the

year (~40%) (solid grey line). Similar pattern can be seen at 40m depth with an increase in surface occurring later (mid-June)
and peaking mid-October at ~60% to drop to ~30% by the end of the year (dashed grey line). Again, these similar features
between the surface and 40m depth happen with a MLD of about 20m, suggesting that the MHW propagates below the MLD
also in this region (Figure 3d, orange line).
At 156m depth – corresponding to the maximum intensity in the mean profile –, unlike for the other two regions, intensity
levels are higher than levels reached for shallower depths (dotted red line). The intensity remains stable throughout the year,
ranging between 0.6°C and 0.8°C. It is higher than the intensity at shallower depth, except for May and October when surface
MHWs develop. High levels of intensity are however not widespread across the subregion as the surface exposed to MHWs
remains around 20% across the year (dotted grey line). Noteworthy, sub-monthly variations are present in the MHW intensity
timeseries suggestive of advective transient features like eddies crossing the domain.

**MHW westward and vertical evolution**
Analysis of MHW within the 3 subregions of the NA, suggests that MHW surface signature propagates westward, and at depth.
To further investigate such dynamics and potential drivers, a 3-dimensional decomposition along longitude, depth and time of
the MHW intensity field and its possible drivers is carried out. The evolution of MHW across the year and the studied regions
is highlighted using Hovmöller diagrams of latitudinal averaged intensity over the 3 subregions at the 3 depths of maximum
intensity (surface, 40m and 156m) (see Methods, Figure 4 a, b and c).
For the surface, the strongest intensity (greater than 0.5°C) takes place primarily in the eastern half of the region (between
60°W and 10°W) and during the months of May to December (Figure 4a). This surface signature of the MHW can be directly
associated with atmospheric features as large positive air temperature anomalies are observed which coincide in time and space
with the MHW intensity patterns (Figure 4d). This suggests a direct response of the surface ocean to the atmospheric anomaly.
The eastern part is characterised by a larger number of peaks from March to December (as seen in the MHW intensity time
series for the NASE subregion Figure 3b), whereas moving westwards to the central part of the region, the period of high
intensity is reduced to the July to October period forming a single large spatiotemporal peak. Furthermore, we note that the
pronounced intensity patterns in the eastern part (anomalies larger than 0.75°C) propagate rapidly westward, most notably
between 10°W and 70°W at an estimated velocity of ~11m/s (first order estimations based on the slope of the intensity pattern
in Figure 4a), starting in July and occurring nearly every month. To the west (70°W to 100°W), fast west propagation from
signal in the central part of the basin can be observed in October. Further west than 80°W a period of strong MHW intensity
(July to October) coincides with a period of strong positive air temperature anomalies.

The patterns of intensity at 40m depth relate strongly with patterns at the surface, namely large intensity in the Eastern half of
the region spanning from April to December (Figure 4b). Peaks in intensity are smaller than for the surface, and patterns
contain less high frequency signal. Similarly to the surface, multiple peaks in intensity characterise the eastern part and a single
long event the central one (30°W -50°W).

The similarity of the MHW signature at 40m with the surface suggests that the atmospheric-driven MHW at the surface also reaches deeper layers. This correlation is confirmed by a depth/time Hovmöller of the NASE region (Figure 4e). Across the period from March to November, the region is exposed to several high peaks in MHW intensity at the surface (as seen in Figure 3b). These propagate rapidly across the mixed layer which vary from 100m to 20m in depth between winter and summer seasons.

This vertical propagation also extends below the MLD (Figure 4e). MHW Intensity larger than 0.5°C can be observed below the MLD from April onwards: at 70m depth in April and progressively reaching 100m by November. The propagation across the mixed layer is rapid, with an estimated velocity of ~4 m/day. Below the MLD, the propagation is slower ranging between 0.7-1.3 m/day. A direct consequence is that the MHW-driven heat accumulation is trapped below the MLD to remain within the ocean interior and be advected far away from the formation area.

At 156m depth, patterns in MHW intensity are very different to what is observed at the surface (Figure 4c). On the eastern side, there is no clear signature with only low MHW intensity levels. To the west, MHW intensity also differs with the surface, but unlike the east some small spatial scale patterns emerge: west of 70°W and for the period of September to November, intensity displays diagonal patterns showing MHW intensity propagating westward with an estimated velocity of ~0.1.m/s. Such westward velocity is characteristic of eddies crossing the Caribbean Basin (Richardson, 2005; Cailleau et al., 2024) suggesting such features are responsible for the intense MHW conditions locally as they trap and carry westward abnormally warm waters. A snapshot of the MHW intensity on the 7th July 2023 at 156m depth overlaid with the Sea Surface Heights anomaly confirms the intensities are trapped in the anticyclonic eddies at depth in this region (Figure 4f). Note that blank areas represent areas where no MHWs were detected or that are outside the studied area (e.g. in the Pacific Ocean).
These very strong local intensities (larger than 2 °C above the 90[th] percentile threshold) are limited in space, and explain the low and stable horizontal extent of MHWs in the CARB region at 156m depth (Figure 3d, dotted grey line). Part of such anomalies come from the NATR region, but predominantly from the Equatorial and North Brazil currents located below the NATR region (e.g. the eddies located at 57°W-10°N on Figure 4f). A detailed study of this region is necessary to understand the processes leading to eddy-trapped heat crossing the region; this however falls beyond the range of our study area and would also require a longer study period spanning beyond 2023 as MHW signatures are still strongly present in 2024 at the equator.

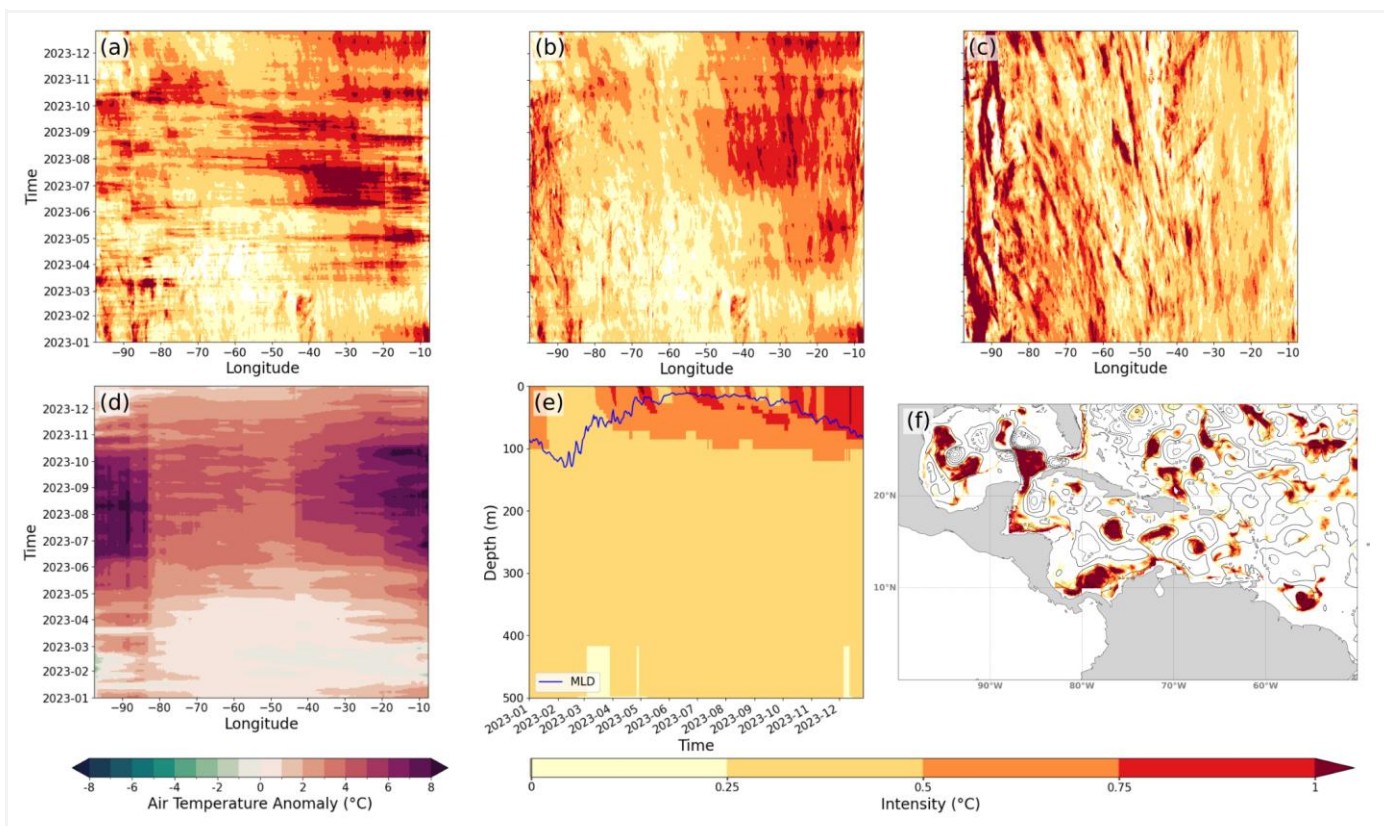

Figure 4: Horizontal MHW and atmospheric characteristics evolution in NASE, NATR and CARB regions. Hovmöller diagrams of MHW intensity at the surface (panel a), at 40m (panel b) and at 156m (panel c) (data averaged over latitude) ; Hovmöller diagram of daily air temperature anomaly at 2m smoothed over a 7-day window (panel d) ; depth/time Hovmöller diagram of intensity in NASE region (panel e) (date averaged over latitude and longitude), the blue line represents the average MLD over the region ; map of MHWs intensity on 07 July 2023 at 156m depth with SSH anomalies contour in black (panel f).


## 4 Discussion and conclusions


Various meteorological and oceanographic estimates showed that the year 2023 was exceptional in terms of heat records, and
in particular the NA region. We studied the region using the Copernicus marine global reanalysis product and characterised
the MHW signature both at the surface and at depth. Compared to previous years, we show the exceptional nature of the 2023
MHW in the NA tropical Ocean which surpasses the last 30 years in terms of duration, intensity and coverage. A strong link
with surface atmospheric conditions is shown (air temperature, negative trade wind anomaly). We also note an evolution of
the timing of MHW maxima during the year with maxima in the East of the basin during the months of May and June, the
central part in mid-summer and the Caribbean Sea in September. A decomposition into different regions of interest for marine
biology (Longhurst provinces), and an in-depth study on these regions highlighted the vertical structure and evolution of
MHWs in each region. We note a progressive penetration of the MHW below the MLD in the Eastern part, together with a

progressive intensification of the MHW intensity across the year. This is a remarkable phenomenon which can be potentially important because it induces a transport into the ocean interior of heat anomalies following surface extreme events. In the West, the Caribbean Sea region shows a very strong MHW signal in the subsurface yet very localised, with a maximum around 156m. These anomalies characteristic of heat trapping eddies originate partly from the NA tropical Ocean but mainly from the North Brazil Current. A dedicated study on eddy-trapped heat pathways to the ocean interior should be considered in the future but will have to cover beyond the year 2023 because in the tropical zone (2°S-2°N) MHWs are still ubiquitous in 2024.

Also, a more comprehensive and detailed quantification of the different contributions of ocean and atmospheric processes is needed to thoroughly understand this unprecedented event. In this sense, Guinaldo et al. (2025) describe the ocean preconditioning and mechanism that lead to the occurrence of this unprecedent event. Also, an approach based on the reconstruction of the heat equation could be done for which the use of the reanalysis would be instrumental to quantify dominant processes (as it provides gridded 3D fields at a 1-day frequency that reduce errors due to the non-linearity of the equation and the approximation of the estimation of the depth of the mixed layer).In view of the exceptional general characteristics of the MHW of 2023 in the NA, further studies are needed, for example to quantify the impact on marine biogeochemistry (BGC), a study for which a BGC reanalysis of Copernicus Marine can be used (GLOBAL_MULTIYEAR_BGC_001_029), but also on the distribution of Sargassum algae – which have a strong societal harmful impact – that develop largely in the Gulf of Guinea and are advected as far as the Caribbean region ( Jouanno et al., 2021). In addition, the definition of extremes could be regionalized and tailored to be representative of harm towards key local species (Capotondi et al., 2024; Oliver et al., 2021).

In this study, the potential of ocean reanalyses to characterise a specific event was shown. Further work on the detection and analysis of extremes would be of interest to assess the MHW impact and importance on the more general climate context. Heat from this NA MHW propagates under the mixed layer to reach different depths depending on the region. Such strong anomalies once away from the surface and trapped within subsurface water masses can potentially be advected over long distances, such as the heat anomalies observed in this study at the equator which were consequently advected to the Caribbean region, the Gulf of Mexico and potentially back into the NA through the Gulf Stream. Detection and monitoring of extremes over the 30 years of the reanalysis will enable to propose an initial scheme and an initial quantification of the importance of such extremes on the overall ocean interior heat content. This estimate will then have to be compared to data sets with a longer time period in order to validate the hypotheses deduced from the study of the GLORYS12 reanalysis fields.

**Acknowledgements**

We would like to thank the Ocean State Report team for the insightful comments and advice in developing of this manuscript

**Author contribution**

SJVG and RBB led the conceptualization of the study, the analysis and writing of the manuscript. AL performed the simulations, data analysis and writing of the manuscript. MD contributed to the conceptualization of the study, and reviewing the manuscript.

**Competing interests**

The authors declare that they have no conflict of interest.

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

**Supplementary materials**

No SP