# Peer review of "The 2023 Marine Heatwave In The North Atlantic Tropical Ocean"

_State of the Planet, 2024_

## Referee Comment (RC1)

Revision of manuscript **sp-2024-31**

This paper investigates the horizontal and vertical intensity and propagation of the 2023 MHW in the North Atlantic upper water column. While the study contains some interesting findings, the lack of clarity and rigour in the explanations and interpretations detracts from its overall impact. The manuscript is difficult to read and understand in several parts.

The Methods section needs to be more rigorously worded, and all calculations listed in the manuscript need to be explained.

The Results need to be improved and appropriate references to figures should be made at the appropriate points in the text to facilitate understanding and strengthen the link between the text and Figures.

The paragraph "Characterisation of Marine Heatwave" should be revised to clearly explain the rationale for the methodological choices and how these choices improve the plausibility and reliability of the results. Providing this context will not only improve readability but also enhance the scientific credibility of the paper.

Specific Comments:

Line 47: "MOI weekly bulletin", add a link or a reference.

Line 51: add references to justify the sentence: "Furthermore, MHW have been well studied for the surface where long satellite records exist, but description and understanding 51 of their vertical structure remains incomplete."

Line 52: add references to Juza et al. (2022) and Pirro et al. (2024)

Juza M, Fernández-Mora À and Tintoré J (2022) Sub-Regional Marine Heat Waves in the Mediterranean Sea From Observations: Long-Term Surface Changes, Sub-Surface and Coastal Responses. Front. Mar. Sci. 9:785771. doi: 10.3389/fmars.2022.785771

Pirro, A., Martellucci, R., Gallo, A., Kubin, E., Mauri, E., Juza, M., Notarstefano, G., Pacciaroni, M., Bussani, A., and Menna, M.: Subsurface warming derived from Argo floats during the 2022 Mediterranean marine heat wave, in: 8th edition of the Copernicus Ocean State Report (OSR8), edited by: von Schuckmann, K., Moreira, L., Grégoire, M., Marcos, M., Staneva, J., Brasseur, P., Garric, G., Lionello, P., Karstensen, J., and Neukermans, G., Copernicus Publications, State Planet, 4-osr8, 18, https://doi.org/10.5194/sp-4-osr8-18-2024, 2024.

Lines 52-58, Pag 2: I take a different view of this statement. For example, Juza et al. (2022) and Pirro et al. (2024) have successfully used SeaDataNet climatology to derive anomalies from Argo float profiles without encountering problems related to 'incomplete reconstruction'. Could you please elaborate on this point and provide additional explanation? In particular, what factors lead to the conclusion that modelling products are more suitable for defining and detecting MHWs than in-situ data?

To rigorously evaluate this claim, a detailed comparison between the 3D model output and in-situ data during a well-documented MHW event is essential. Such a study would assess the ability of the model to reproduce the observed trends, particularly in terms of intensity, duration and spatial variability. A key question is whether the model accurately represents the observed dynamics or whether it over-smooths the data, potentially underestimating localised extremes.

Pag 4, Lines 85-90: Which layer did you use to define the occurrence of MHW and apply the method of Hobday et al. 2026? Did you use only the first layer of the model (surface layer) or the 0-200 m layer? Please clarify.

Pag 4, Lines 90-92: This sentence is rather unclear and raises questions about the authors' methodology and aims. Why did the authors estimate the MHW for the entire water column in 2023, but limit their analysis to the surface layer for the period 1993–2022? What was the purpose of these different approaches? The reasons for these estimates are not clearly explained, leaving the reader uncertain about the authors' goals and the reasons for their choices.

Introduction and Line 35, Pag:5: There was also a marine heatwave in 2021 and 2022 in the North Atlantic and the Mediterranean (e.g. https://www.mercator-ocean.eu/en/news/state-of-the-climate-in-europe-2022-report-2/). Include these events in the introduction and relate it to that of 2023. Could it be the occurrence of these earlier events that intensified the MHW of 2023?

Figure 3a: The difference in colour between NASE (brown) and CARB (red) is too difficult to see in the legend. Why not use less similar colours?

Pag 8, Line 78: There is also a peck (absolute maximum) in March in the CRAB region but no comment on it in the text.

Pag 9, Lines 81-94: This paragraph is very difficult to follow for a number of reasons: the relative figures are not quoted in the text and/or are quoted incorrectly (this applies to the entire results section); the depths of the layers quoted in the text do not coincide with those in Figure 3 (e.g. 100 m in the text and 156 m in the figure); the references to MLD are incomprehensible as the MLD is not shown in any of the figures.

Pag 9, Line 96: "The evolution of the mean intensity for NATR describes..." In which layer?

Pag 10, Lines 22-38: If this map was created by averaging the selected areas in latitude, how did you manage the overlap in longitude of the NASE and NATR sections? If I look east of 40°W, am I looking at NATR, NASE or both?

Pages 10-11: How were all the velocities estimated on these pages (lines 36, 51, 58) There is no reference to velocities in the methods.

Figure 4f: What is the meaning of the blank areas in Figure 4f? Is it the lack of data? Why is it that when you use a model there is no data in some areas? Could you please describe this figure better, explain how it was made and how it should be interpreted?

---

## Author Comment (AC1)

Revision of manuscript **sp-2024-31**

This paper investigates the horizontal and vertical intensity and propagation of the 2023 MHW in the North Atlantic upper water column. While the study contains some interesting findings, the lack of clarity and rigour in the explanations and interpretations detracts from its overall impact. The manuscript is difficult to read and understand in several parts.

The Methods section needs to be more rigorously worded, and all calculations listed in the manuscript need to be explained.

The Results need to be improved and appropriate references to figures should be made at the appropriate points in the text to facilitate understanding and strengthen the link between the text and Figures.

The paragraph "Characterisation of Marine Heatwave" should be revised to clearly explain the rationale for the methodological choices and how these choices improve the plausibility and reliability of the results. Providing this context will not only improve readability but also enhance the scientific credibility of the paper.

We thank the reviewer for the very useful feedback to improve the manuscript. We have taken them into account to enhance clarity and rigor. As requested, the method section was fully rewritten to better express our goal. Figure were better quoted in the results section to improve the readability and understandings.

Specific Comments:

Line 47: "MOI weekly bulletin", add a link or a reference.

Link was added in the introduction.

Line 51: add references to justify the sentence: "Furthermore, MHW have been well studied for the surface where long satellite records exist, but description and understanding 51 of their vertical structure remains incomplete."

The sentence has been modified and now states that 'the subsurface extent should be considered more in details' to better reflect the current state of the art. We also added the following references to Zhang et al. 2023, Schaeffer et al. 2023 and Sun et al. 2023 in the revised manuscript to support this.

Zhang, Y., Du, Y., Feng, M., and Hobday, A. J.: Vertical structures of marine heatwaves, Nat Commun, 14, 6483, https://doi.org/10.1038/s41467-023-42219-0, 2023

Schaeffer, A., Sen Gupta, A., and Roughan, M.: Seasonal stratification and complex local dynamics control the sub-surface structure of marine heatwaves in Eastern Australian coastal waters, Commun Earth Environ, 4, 1–12, https://doi.org/10.1038/s43247-023-00966-4, 2023.

Sun, D., Li, F., Jing, Z., Hu, S., and Zhang, B.: Frequent marine heatwaves hidden below the surface of the global ocean, Nat. Geosci., 16, 1099–1104, https://doi.org/10.1038/s41561-023-01325-w, 2023.

Line 52: add references to Juza et al. (2022) and Pirro et al. (2024)

Juza M, Fernández-Mora À and Tintoré J (2022) Sub-Regional Marine Heat Waves in the Mediterranean

Sea From Observations: Long-Term Surface Changes, Sub-Surface and Coastal Responses. Front. Mar. Sci. 9:785771. doi: 10.3389/fmars.2022.785771

Pirro, A., Martellucci, R., Gallo, A., Kubin, E., Mauri, E., Juza, M., Notarstefano, G., Pacciaroni, M., Bussani, A., and Menna, M.: Subsurface warming derived from Argo floats during the 2022 Mediterranean marine heat wave, in: 8th edition of the Copernicus Ocean State Report (OSR8), edited by: von Schuckmann, K., Moreira, L., Grégoire, M., Marcos, M., Staneva, J., Brasseur, P., Garric, G., Lionello, P., Karstensen, J., and Neukermans, G., Copernicus Publications, State Planet, 4-osr8, 18, https://doi.org/10.5194/sp-4-osr8-18-2024, 2024.

We thank the reviewer for pointing to these valuable studies and added both references intpo the revised manuscript

Lines 52-58, Pag 2: I take a different view of this statement. For example, Juza et al. (2022) and Pirro et al. (2024) have successfully used SeaDataNet climatology to derive anomalies from Argo float profiles without encountering problems related to 'incomplete reconstruction'. Could you please elaborate on this point and provide additional explanation? In particular, what factors lead to the conclusion that modelling products are more suitable for defining and detecting MHWs than in-situ data?

To rigorously evaluate this claim, a detailed comparison between the 3D model output and in-situ data during a well-documented MHW event is essential. Such a study would assess the ability of the model to reproduce the observed trends, particularly in terms of intensity, duration and spatial variability. A key question is whether the model accurately represents the observed dynamics or whether it oversmooths the data, potentially underestimating localised extremes.

We thank the reviewer for his/her comment, we have removed the sentence which appeared to oppose approaches and was misleading.

We believe both approaches (use of data-assimilating modelling products and in-situ data) are complementary, we chose to use modelling-based products to be able to build a daily 30-year climatology at surface and subsurface and for every grid cell. This choice is mainly motivated by the recommendations of the WMO to use a 30-year climatology when possible and the recommendations of Hobday et at. 2016, 2018 (used for MHW definition). Also, it is important to keep in mind that we didn't use a model product (e.g. a free run model resulting from solely resolving the equations of state) but a reanalysis one which is constrained by data (in situ and satellite-driven), and as such is close to the observation derived data.

Pag 4, Lines 85-90: Which layer did you use to define the occurrence of MHW and apply the method of Hobday et al. 2026? Did you use only the first layer of the model (surface layer) or the 0-200 m layer? Please clarify.

The method of Hobday et al. 2016 was used for the surface layer of GLORYS12V1 reanalysis (thickness of 1m) from 1993 to 2023 and used for all GLORYS12V1 layers from surface to 2,200m for the year 2023. Sentences were added to clarify this aspect lines 90-93.

Pag 4, Lines 90-92: This sentence is rather unclear and raises questions about the authors' methodology and aims. Why did the authors estimate the MHW for the entire water column in 2023, but limit their analysis to the surface layer for the period 1993–2022? What was the purpose of these different approaches? The reasons for these estimates are not clearly explained, leaving the reader uncertain about the authors' goals and the reasons for their choices.

We first detected MHWs at the surface for 2023 and also from 1993 to 2022 to compare the 2023 surface characteristics with those of previous year. This first part allows us to claim the exceptional 2023 event for the surface. Once the surface 2023 event was characterized, we extended the study to subsurface to understand the mechanisms.

Method section was fully rewritten for clarification and further explanations.

Introduction and Line 35, Pag:5: There was also a marine heatwave in 2021 and 2022 in the North Atlantic and the Mediterranean (e.g. https://www.mercator-ocean.eu/en/news/state-of-the-climate-inhttps://www.mercator-ocean.eu/en/news/state-of-the-climate-in-europe-2022-report-2/europe-2022-report-2/). Include these events in the introduction and relate it to that of 2023. Could it be the occurrence of these earlier events that intensified the MHW of 2023?

This previous events could have indeed played a role in the intensity of the 2023 event. Nevertheless, it is complicated to disentangle the initial condition and atmospheric influence with only one occurrence. Moreover, we estimated the start of the event in spring (Figure 1b) and we distinguish peaks and increase in intensity as from March in the time series (Figure 3 b, c & d), making this event and the ones from 2020 and 2021 separate ones. This study focuses on the 2023 event and its characteristics.

Even thought, we acknowledge your claim that these previous MHW from 2020 and 2021 might have played a role in the 2023 event, we chose not to mention them to avoid any confusion for the reader as we don't analyze this potential role later in the study.

Figure 3a: The difference in colour between NASE (brown) and CARB (red) is too difficult to see in the legend. Why not use less similar colours?

Colors were changed to better differentiate NASE & CARB.

Pag 8, Line 78: There is also a peck (absolute maximum) in March in the CRAB region but no comment on it in the text.

This peak in March was mentioned later in the result section. We now mention it earlier and added an extra sentence in the revised manuscript. It seems to come from MHWs trapped by the loop current in the Gulf of Mexico that peak mid-March and quickly disappear (see intensity maps below). This is an independent event which would require another study.

[Figure]

MHW Intensity map 2023-03-03 (a), 2023-03-13 (b), 2023-03-23 (c)

Pag 9, Lines 81-94: This paragraph is very difficult to follow for a number of reasons: the relative figures are not quoted in the text and/or are quoted incorrectly (this applies to the entire results section); the depths of the layers quoted in the text do not coincide with those in Figure 3 (e.g. 100 m in the text and 156 m in the figure); the references to MLD are incomprehensible as the MLD is not shown in any of the figures.

The quotes were corrected for the entire results section.

The quoted depths correspond to the layer in the figure. We mentioned 100m which is not shown in Figure 3 but we can see in the time/depth Hovmöller (figure 4e) which is quoted.

The MLD is shown in Figure 3a and Figure 4e and we added its temporal evolution in each subregion in Figure 3b, c & d. We added a sentence in the revised method section and in figure description for clarifications. We also expended comments on the MLD in the results sections.

Pag 9, Line 96: "The evolution of the mean intensity for NATR describes…" In which layer?

This sentence refers to the evolution throughout the water column we can see on Figure 3a (intensity profile), and especially for the layers shown in the time series of Figure 3c. We added 'at the surface and at depth' for clarification.

Pag 10, Lines 22-38: If this map was created by averaging the selected areas in latitude, how did you manage the overlap in longitude of the NASE and NATR sections? If I look east of 40°W, am I looking at NATR, NASE or both?

Thank the reviewer for highlighting the lack of clarity on this point. When sections overlap in longitude, the data from both sections are averaged together. We specified this aspect when rewriting the revised method section.

Pages 10-11: How were all the velocities estimated on these pages (lines 36, 51, 58) There is no reference to velocities in the methods.

Velocities were roughly estimated based on the slope of diagonals formed by intensity in Hovmöller diagrams. This provides an estimate of the order of magnitude of the velocities, but it is not intended to represent an exact value. Explanations were added in the revised manuscript.

Figure 4f: What is the meaning of the blank areas in Figure 4f? Is it the lack of data? Why is it that when you use a model there is no data in some areas? Could you please describe this figure better, explain how it was made and how it should be interpreted?

Figure 4f represents the intensity of MHWs at 156m depth on the 7th of July 2023, thus it only shows areas where MHW were detected in the Atlantic Ocean. Blank areas mean that no MHW were detected there for that day. The Pacific Ocean is blank as well as it is out of the studied area. A sentence was added in the revised manuscript to clarify this aspect.

---

## Author Comment (AC2)

**Review of "The 2023 Marine Heatwave In The North Atlantic Tropical ocean" by Loubet et al.**

This work presents an analysis of the 2023 marine heatwaves that happened in the North Atlantic Ocean in 2023, an unprecedented year in terms of temperature anomaly and extreme values. The work is timely and interesting, although I have the impression it lacks some depth in the discussion of the results presented, which may be imposed by the lengths of the works submitted to this issue. This makes the manuscript superficial at times, and we have the impression the authors rush over some of the topics.

My major comment would be for the authors to include more detailed information (including a figure) on the spatio-temporal distribution of the mixed layer depth, which is an important parameter in the sub-surface propagation of marine heatwaves and its role depends on the region.

We thank the reviewer for his constructive feedback. We agree that some aspects could be investigated further, but the scope of the OSR encouraged us to maintain a concise and focused approach, which led us to prioritize key findings and highlight only the main points. We nonetheless mentioned in the 'Discussion and conclusions' section some topic for further study such as the origin and mechanism of the eddies in the CARB region, the biological impact of the event or the impact of the heat trapped in the ocean interior below the mixed layer.

We acknowledge the importance of the mixed layer depth (MLD) in subsurface propagation of MHW and recognize that it is not detailed extensively in the paper and is explained by the limited number of figures in the OSR format, and priority was given for other results we found more relevant to characterize the 2023 MHW in the North Atlantic Tropical Ocean. We only show the MLD for the NASE regions (figure 4e) to illustrate its temporal evolution relative to the temporal evolution of the MHW in the vertical for the region. We have now added the mean MLD temporal evolution for each region of focus in figure 3 (b, c and d). We added further insights on the link between MLD and MHW intensity signature at depth in the revised manuscript. We also mentioned the MLD in the section 'MHW westward and vertical evolution' highlighting the MHW propagation below the MLD (lines 282 to 286).

In this respond, you will also find maps of monthly mean MLD anomalies for further details into its spatial evolution (see below as a respond to a specific comment).

Please find hereafter our responses for the reviewer's specific comments, in order of appearance.

- Abstract: It is mentioned that the MHW peaks in June, but in the short summary after the abstract it says May.

Thank you for highlighting this mistake, it has now been fixed in the abstract.

- Line 51. I have seen the claim that subsurface MHWs are less studied multiple times, but the field of MHW is being studied by a large amount of scientists, and subsurface MHWs have received a lot of attention as well. The authors themselves provide a good list of references about subsurface MHWs. I would rephrase this or remove this sentence.

We agree with the reviewer that subsurface MHWs have recently received a lot of attention. We have rephrased the sentence to focus more on the need to study subsurface MHW in details and added further references (Schaefer et al., 2023; Zhang et al., 2023; Sun et al., 2023).

- Line 56: "but require the system…" I don't understand the "but" here, it looks like you are presenting reanalysis as negative because they require all these things (data assimilation, high spatial and temporal resolution…) but you are going to use a reanalysis, so why presenting it as if it was a hindrance?. Many reanalysis assimilate data, and have the necessary temporal coverage Also, in order to study open waters MHW, the requirement of high spatial resolution is not that crucial, and most reanalysis would have a sufficient resolution. In fact it would be nice to know why your choice of reanalysis (GLORYS 12V1 from CMEMS) is the adequate with respect to others.

We agree that the word 'but' was not the correct one to choose, we wanted to highlight the requirements needed when using a reanalysis product in order to study MHW.

As you mentioned, high spatial resolution is not necessarily a requirement to study MHWs in open water. However, our study area is wide and contained region of different nature. We highlighted in the paper the important role of eddies in heat propagation in the Caribbean region, leading to subsurface MHWs. Thus, a high resolution and eddy resolving system, such as GLORYS12V1, was here necessary.

Furthermore, we used GLORYS12V1 because it has been intensively validated for many regions of the ocean (as detailed in Amaya et al., 2023). GLORYS12V1 has also have been used in subsurface MHW studies (Sun et al., 2023 ; Fernández-Barba et al., 2024).

Fernández-Barba, M., Huertas, I. E., and Navarro, G.: Assessment of surface and bottom marine heatwaves along the Spanish coast, Ocean Modelling, 190, 102399, https://doi.org/10.1016/j.ocemod.2024.102399, 2024.

Amaya, D. J., Jacox, M. G., Alexander, M. A., Scott, J. D., Deser, C., Capotondi, A., and Phillips, A. S.: Bottom marine heatwaves along the continental shelves of North America, Nat Commun, 14, 1038, https://doi.org/10.1038/s41467-023-36567-0, 2023.

- Line 94: some symbols are missing in the formula so I cannot assess its correctness

We apologise for the confusion. The issue was not missing symbols but blank spaces that appeared due to conversion between different document types. It has now been fixed.

- Figure 1. The lines used in panels b are too similar in colour (I printed the paper to read it and I had to revert to screen for the images, but even so I found them too similar). Also, on panel c, you choose to colour events by mean activity, but the events display an increasing Activity with increasing duration and intensity (x and y axes), so first come all greys, then the greens, etc, so not much newer information is shown. Maybe colouring by decade would give us information of which years are anomalous in the progressive increasing trend? For example, 1998 would stand as a anomalous year for its decade.

Concerning panel b, as each coloured line represent the extent of a particular MHW category (moderate, strong severe and extreme), the colours chosen are the ones commonly used to refer to these MHW category (yellow for moderate, orange for strong, bright red for severe and dark red for extreme) in the literature and in this document consistently across figures (figure 1b and 2a). Nevertheless, to address the reviewer's comment and improve the differentiation between colors, we increased the thickness of the lines.

Concerning panel c, we acknowledge the position and size of a bubble/year and its colour show twice the same information. Nevertheless, this plot was inspired by published figures in the literature used to characterize MHW events (Darmaraki et al., 2019).

It was opted to use the activity as the parameter for the color of the bubbles as it is the variable that encompasses the contribution of all other chatacteristics (extent, duration, intensity), and enables to classify the severity of MHW for each year, and highlights how exceptional 2023 was. In our opinion more information is conveyed with such colorstyle (rather than a color per decade), for instance it shows 2010 and 2020 are the 2[nd] and 3[rd] year in terms of MHW activity.

To address the reviewer's comment, we generated a plot with colour representing decades (see figure below). It indeed shows the warming across decades. Such coloring

emphasizes the signal of climate change, however as the focus here is for a particular year we find that using colouring for the activity better suited.

[Figure]

Same plot as figure 1c but colours represent decades: grey (1993-1999), green (2000-2009), blue (2010-2019) and red (2020-2023)

 Figure 2: too many colour in panel b, I cannot really see anything... I would colour by season form example, or at least use the same palette for each season (shades of blue for winter, shades of orange in fall... But I think one colour per season should be more than enough.

We followed the reviewer's recommendation and changed colours making the figure easier to interpret. We toned down the colours and used only one shade per season which is enough to see the westward evolution (see figure below). We also tried using one colour per season but found that using different colours by month illustrated better the evolution from east to west, particularly in the NATR region (orange shades).

[Figure]

Same plot as figure 2b

Line 72: "Evolution of MHW Intensity and Extent" add "at depth" to the title?

We thank the reviewer for this advice, we added 'across depth' to the title.

Line 78: May (not mai)

Corrected, we thank the reviewer for highlighting this mistake.

Figure 3. Again the colour here are too similar and in panel a we cannot really see the difference between NASE and CARB. In panels b, c and d, consider using the same colours for the same depth (i.e. 40 m, continuous orange for mean intensity, dashed orange for spatial extent (not extend as in the legend). Also in the caption these lines are referred to as black, but they are grey.

Adding colour for the spatial extent would make the figure confusing as the extent at the surface is also a solid line. To emphasize the link extent/intensity we chose to use the same linestyle for each depth (solid for surface, dashed for 40m and dotted for 156m). This also allowed us to use one colour per region leading to a stronger link between the left plot and the rigth ones. We also were able to use other colours in the panel a improving the distinction between regions.

To highlight the role of the mixed layer in MHW propagation at depth (and address a general comment of the reviewer), the MLD temporal evolution was added for each subregion (orange lines).

[Figure]

Figure 3: Evolution of the intensity and surface coverage of the 2023 marine heatwave for different regions in the North Atlantic. Mean MHW intensity (in °C) profile (panel a) of NASE (blue), NATR (green) and CARB (red). Shading areas represent standard deviations of spatial mean. Dotted horizontal lines represent the mean MLD and red dots represent depth of highest mean intensity for each region. Time series of mean intensity (in °C), surface coverage (in %) and mean mixed layer depth (in m) in NASE (panel b), NATR (panel c) and CARB (panel d) provinces. Blue, green and red lines represent the mean intensity at the surface (solid), at 40m (dashed) and at 150m (dotted). Grey lines represent the surface coverage at the surface (solid), at 40m (dashed) and at 156m (dotted). Orange lines represent the mean layer depth in the corresponding region.

Line 85. The sentence starting "It suggests..." is quite complex, please separate it in two or reformulate.

We acknowledge that the sentence is confusing. We rephrased it to improve clarity in the revised manuscript: The MHW signature at the surface develops earlier than at subsurface: it suggests that the signal propagates from the surface, across the water column, to subsurface.

In this section (starting in line 73) is where the MLD starts to be mentioned and referred to, to discuss the results. I think a figure showing the depth of the mixed layer at the different regions discussed is missing, for different times of year. It would be much easier to understand some of the results shown.

We thank the reviewer for his suggestion. It is true that a more thorough study of the MLD would be interesting. However, we should not exceed 4 figures to respect OSR format. Nevertheless, we partially addressed this by adding the temporal evolution of the mean mixed layer depth over the 3 subregions studied in the figure 3 (orange lines) and further explanations in the revised text

We plotted maps of the monthly mean MLD anomalies to investigate its spatial evolution (see plots below). The MLD anomaly is very negative in the NASE region in March (beginning of the event) which means that the MLD was shallower than normal. This is expected as a result of the warming caused by atmospheric phenomena which is coherent with our findings.

In NASE, the surface intensity peaks at the very beginning of May (Figure 3b). The period of increase in intensity leading to this peak is in April, a month during which the MLD anomaly is mostly negative over the region (meaning the MLD is shallower than usual), which is coherent with a surface warming (not shown).

In May, the mean MLD anomaly is mostly positive in the NASE region (meaning the MLD is deeper than usual), and is concomitant with a decrease of the MHW intensity at the surface, (figure 3b).

In June, the mean MLD anomaly is mostly negative, particularly in the NATR region (meaning the MLD is shallower than normal). This is, again, coherent with the peak of surface intensity observed in the NATR region during the month of June (Figure 3c).

In October – month of surface intensity peak in CARB – the MLD anomalies is mostly negative in CARB. More specifically, the variation of the MLD follows the variations of the intensity, meaning that the MLD shallows when the surface intensity increases (Figure 3d).

[Figure]

Map of monthly mean MLD Anomaly. March (top left), May (top rigth), June (bottom left), october (bottom rigth). Zones delimited in black refer to the Longhurst provinces of focused: NASE (east), NATR (middle) and CARB (west).

Line 117 (the pdf is cut and all numbers after 100 appear with only 2 digits, so line 17 in fact): "is always higher than shallower depth". I don't understand this sentence.

The sentence is indeed unclear. We rephrased it : 'The intensity remains stable throughout the year, ranging between 0.6 °C and 0.8 °C. It is higher than the intensity at shallower depth, except for May and October when surface MHWs develop.'

Here, we wanted to emphasise the higher intensity at depth (156m, red dotted lines) compared to the surface (red solid line) and 40m (red dashed line) (figure 3).

Figure 4. Caption shows "copyright" symbol instead of (c). First mention to panel (f) should be panel (d). In panel d there is a vertical line at about 80degrees west, what is causing this? I was surprised to see that scales seem to be finer and shorter at 156 m than at the surface, what is causing this? I guess this is the eddies mentioned in the paper that transfer heat to other latitudes, but why are these eddies forming at this depth (and not at others).

The copyright symbol has been replaced with a `c` in the caption and the issues with figure quoting have been fixed.

Panel d shows indeed a meridional discontinuity at 85°W (Gulf of Mexico). This can be explained by the data being averaged over the latitude direction. As the region further west than 85°W becomes narrower and only contains a semi-enclosed oceanic region which is overall warmer,  it creates the visual effect of a jump in average temperature going westward.

The scales of patterns are indeed thinner at 156m.The main signal comes from oceanic processes (which are small in scale than atmospheric processes) whereas the surface is more driven by atmospheric phenomenon which are larger in scale (note the first Rossby radius difference between ocean and atmosphere).  As mentioned by the reviewer, we believe the signature comes from eddies and as we explained in the manuscript they seem to have originated from other regions like North Brazil currents and are transported into the region.
We focus our study at 156m depth because it is where we have observed the maximum of MHW intensity in the CARB region (figure 3). We found other similar features at different depth (100m for example, see figure below). However, a detailed study of this phenomena is necessary to understand the exact processes and the involved regions but such investigation falls is beyond the scope of our study area and would require a longer study period.

[Figure]

Hovmöller diagrams of MHW intensity at 100m depth (data averaged over latitude) (left) ; map of MHWs intensity on 2023-07-07 at 100m depth with SSH anomalies contour in black (rigth).